# Effect of sub-bandgap defects on radiative and non-radiative open-circuit voltage losses in perovskite solar cells

Guus J. W. Aalbers [1,3], Tom P. A. van der Pol [1,3], Kunal Datta [1], Willemijn H. M. Remmerswaal[1], Martijn M. Wienk[1] & René A. J. Janssen [1,2] ✉

The efficiency of perovskite solar cells is affected by open-circuit voltage losses due to radiative and non-radiative charge recombination. When estimated using sensitive photocurrent measurements that cover the above- and sub-bandgap regions, the radiative open-circuit voltage is often unphysically low. Here we report sensitive photocurrent and electroluminescence spectroscopy to probe radiative recombination at sub-bandgap defects in wide-bandgap mixed-halide lead perovskite solar cells. The radiative ideality factor associated with the optical transitions increases from 1, above and near the bandgap edge, to ~2 at mid-bandgap. Such photon energy-dependent ideality factor corresponds to a many-diode model. The radiative open-circuit voltage limit derived from this many-diode model enables differentiating between radiative and non-radiative voltage losses. The latter are deconvoluted into contributions from the bulk and interfaces via determining the quasi-Fermi level splitting. The experiments show that while sub-bandgap defects do not contribute to radiative voltage loss, they do affect non-radiative voltage losses.

Metal-halide perovskite solar cells (PSC) are of great interest for high-efficiency multijunction applications. In particular, mixed-halide wide-bandgap perovskites yielding a high open-circuit voltage ($V_{OC}$) are critical to the efficiency of multijunction devices. Currently, such devices suffer from performance losses due to $V_{OC}$ deficits commonly ascribed to the presence of electronic defective states in the perovskite, at grain interfaces, or charge-selective interfaces[1–5]. Identifying electronic defect (trap) states and understanding the effect of such states on the $V_{OC}$ is crucial to reduce losses and enhancing solar cell performance.

Sensitive spectroscopic techniques, such as electroluminescence (EL), photoluminescence (PL), and photocurrent measurements, are commonly used to characterize low-energy (sub-bandgap) states in thin semiconductor films and devices. For instance, PL measurements on perovskite thin films have shown low-energy emissions related to the presence of shallow defects and EL has been used to identify defect states in metal oxide semiconductors[6–8]. Also, highly sensitive photocurrent measurements on perovskite solar cells have revealed sub-bandgap states associated with deep defects[4,5,9]. Frequency domain techniques, such as thermal admittance spectroscopy or drive-level capacitance profiling[10,11], allow for probing both optically active and inactive defect densities and the trap density of states of thin semiconductor films. However, these techniques do not yield insight into the loss mechanisms associated with detected defect states as optical methods such as PL, EL, or photocurrent would provide.

The $V_{OC}$ of any solar cell is limited by the bandgap ($E_g$) of the semiconductor in the active layer and is further reduced by radiative and non-radiative recombination of electrons and holes that are generated after photon absorption. Often, the $V_{OC}$ is expressed by the

[1]Molecular Materials and Nanosystems and Institute for Complex Molecular Systems, Eindhoven University of Technology, P.O. Box 513, 5600 MB Eindhoven, The Netherlands. [2]Dutch Institute for Fundamental Energy Research, De Zaale 20, 5612 AJ Eindhoven, The Netherlands. [3]These authors contributed equally: Guus J. W. Aalbers, Tom P. A. van der Pol. ✉e-mail: r.a.j.janssen@tue.nl

relation[12]:

$$V_{OC} = V_{OC}^{rad} - \Delta V_{OC}^{non-rad} \qquad (1)$$

where $V_{OC}^{rad}$ is the open-circuit voltage in the radiative limit where all recombination is radiative, and $\Delta V_{OC}^{non-rad}$ is the loss in $V_{OC}$ due to non-radiative recombination. Presently, the individual contributions of radiative and non-radiative processes to the voltage deficit ($E_g - qV_{OC}$) are not accurately known. From the Shockley diode equation, the definition of $V_{OC}^{rad}$ follows as[13]:

$$V_{OC}^{rad} = \frac{n_{id}^{rad} k_B T}{q} \ln\left(\frac{J_{ph}}{J_0^{rad}} + 1\right) \qquad (2)$$

where $k_B$ is Boltzmann's constant, $T$ is the absolute temperature of the solar cell, $q$ is the elementary charge, $n_{id}^{rad}$ is the dimensionless radiative ideality factor, $J_{ph}$ is the photocurrent density, and $J_0^{rad}$ is the radiative saturation current density. The photocurrent density and radiative saturation current density are given by[14]:

$$J_{ph} = q \int_0^\infty Q_e^{PV}(E) \varphi_{AM1.5G}(E) \, dE \qquad (3)$$

$$J_0^{rad} = q \int_0^\infty Q_e^{PV}(E) \varphi_{bb}(E) \, dE \qquad (4)$$

where $E$ is the photon energy, $Q_e^{PV}(E)$ is the external quantum efficiency (EQE) of the solar cell, $\varphi_{AM1.5G}(E)$ the photon flux of the incident AM1.5G solar spectrum, and $\varphi_{bb}(E)$ the black-body spectrum photon flux given by:

$$\varphi_{bb}(E) = \frac{1}{4\pi^2 \hbar^3 c^2} \frac{E^2}{\exp\left(\frac{E}{k_B T}\right) - 1} \qquad (5)$$

where $\hbar$ is the reduced Planck constant and $c$ is the speed of light in a vacuum. By inserting Eqs. (3)–(5) into Eq. (2), $V_{OC}^{rad}$ can be calculated, if $n_{id}^{rad}$ is known, using only the EQE of the solar cell.

Despite its simplicity, this method to determine the radiative limit and subsequently non-radiative losses is—in practice—limited by the spectral range of the EQE measurements that is experimentally accessible, which is less than the integration limits ($0 \to \infty$) in Eqs. (3) and (4). Because of the steep rise of the black-body spectrum at lower energies, the lower integration limit has a very strong influence on the value calculated for $J_0^{rad}$. To extend the lower integration limit, experimental studies typically extrapolate the band-edge in EQE spectra[12,15]. Interestingly, sensitive experimental techniques have been developed recently that enable measuring the EQE in the sub-bandgap region[4,16]. At low energies, the sub-bandgap states (often assumed to be charge trapping and referred to as defects/traps) give rise to features diverging from the exponential band-tail. Surprisingly, if the EQE of these sub-bandgap states is taken into account in Eq. (4), the $V_{OC}^{rad}$ calculated from Eq. (2) can be less than the measured $V_{OC}$[16]. To account for this unphysical result, a phenomenological double-diode model has been proposed[16]. This model assigns different ideality factors ($n_{id}$) to bandgap (band-to-band, BTB) transitions ($n_{id,1} = 1$) and sub-bandgap transitions ($n_{id,2} = 2$) that follow Shockley–Read–Hall (SRH) theory. However, describing all sub-bandgap transitions using the ideality factor derived for deep defect states requires further experimental and theoretical justification that is currently unavailable. The ideality factor of defect states should thus be investigated to circumvent unphysical results or generalizations that result from employing single- or double-diode models, respectively. Experimentally, ideality factors of solar cells are commonly determined through light intensity-dependent $V_{OC}$ measurements and generally fall between 1 and 2 for PSCs[17,18].

In this study, the influence of sub-bandgap defect states on the recombination of charge carriers in mixed-halide perovskite solar cells is examined. We identify emissive sub-bandgap states in the EL spectrum and correlate the emission to sub-bandgap photocurrent generation measured through sensitive photocurrent spectroscopy. By relating EL and EQE through optical reciprocity, we obtain photon energy-dependent radiative ideality factors for the sub-bandgap transitions. This allows extending the Shockley diode equation to a many-diode model, which enables determining the radiative voltage limit and quantifying non-radiative voltage losses via Eq. (1) by including the sub-bandgap generated photocurrent in a theoretically consistent manner. In the many-diode model, $V_{OC}^{rad}$ is negligibly influenced by sub-bandgap states. We demonstrate the applicability of the many-diode model by deconvoluting radiative and non-radiative voltage losses in different perovskite solar cells with various bandgaps, with and without surface passivation of the perovskite layer. Using quasi-Fermi level splitting experiments, we disentangle non-radiative losses in contributions from the perovskite bulk and from its interfaces with the individual charge-transport layers. Finally, we show that the non-radiative voltage losses correlate with sub-bandgap photocurrent generation.

## Results

$Cs_{0.05}(FA_{0.83}MA_{0.17})_{0.95}Pb(I_{0.83}Br_{0.17})_3$ triple-cation mixed-halide perovskite (with MA = methylammonium and FA = formamidinium, CsFAMA-17) p-i-n solar cells were fabricated on glass substrates with a patterned indium tin oxide (ITO) front electrode covered with a [2-(9H-carbazol-9-yl)ethyl]phosphonic acid (2PACz) self-assembled hole-transporting monolayer. After depositing the perovskite[19], the device was completed with layers of $C_{60}$ and bathocuproine (BCP), and an Al back electrode (Fig. 1a). The current density vs. voltage (J-V) scan of the champion solar cell shows minimal hysteresis and provides a $V_{OC}$ of 1.10 V, a short-circuit current density ($J_{SC}$) of 19.9 mA cm$^{-2}$, and a fill factor (FF) of 0.79, leading to a power conversion efficiency (PCE) of 17.3% (Fig. 1b). The mean and median $V_{OC}$ determined from 25 devices are 1.07 ± 0.04 V and 1.09 V, respectively (Supplementary Fig. 1). The EQE approaches 90% under 1-sun equivalent bias illumination (Fig. 1b). Integration of the EQE with the AM1.5G spectrum affords a $J_{SC}^{EQE}$ of 20.1 mA cm$^{-2}$ and provides an estimated PCE of 17.5%, in good agreement with values obtained from the J-V scan. The optical bandgap (1.63 eV) was determined by finding $\frac{d^2 Q_e^{PV}}{dE^2} = 0$ (Supplementary Fig. 2)[12].

### Radiative losses

To investigate the sub-bandgap states of these mixed-halide perovskite solar cells, we use sensitive photocurrent measurements and study the EQE below the bandgap. A typical highly sensitive EQE spectrum for a CsFAMA-17 solar cell is shown in Fig. 1c in a semilogarithmic plot as a function of photon energy. This EQE spectrum can be divided into three regions, namely the above-bandgap region, the band-tail, and the sub-bandgap contribution. The above-bandgap region corresponds to the EQE spectrum presented in the inset of Fig. 1b. At energies below the bandgap, an exponential decrease occurs, commonly known as the Urbach tail[20]. This Urbach tail has been ascribed to energetic disorder within the perovskite material and arises due to band tailing of the valence and conduction bands. At even lower energies, the EQE deviates from the Urbach tail and the signals are attributed to sub-bandgap defect states. In this sub-bandgap region, two distinct defects that are typical for mixed-halide perovskites are present[4,21]. We note that the shape of the EQE spectrum in this region is strongly influenced by optical interference effects as a result of which the true energetic distribution of sub-bandgap states differs from the measured EQE[4].

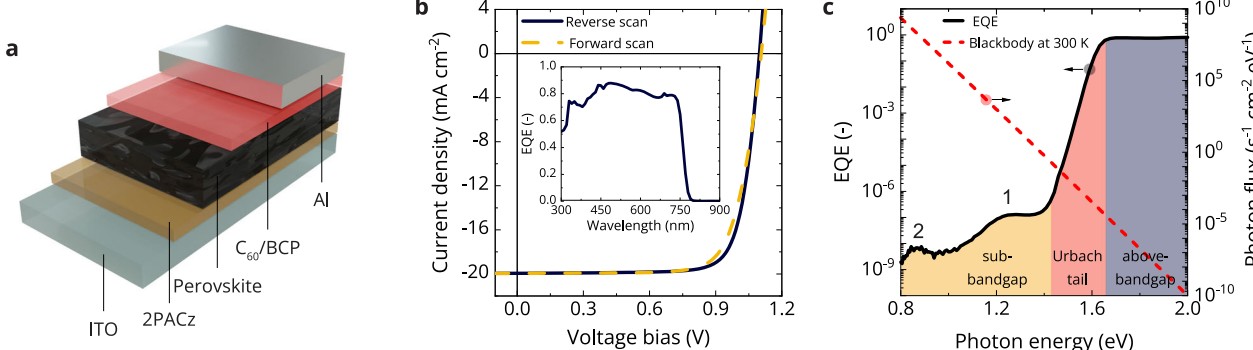

**Fig. 1 | Device performance and sub-bandgap response in the p-i-n perovskite solar cells. a** Perovskite p-i-n device architecture used in this work. **b** Current density–voltage (*J-V*) measurements in reverse (solid line) and forward (dashed line) scan directions of ITO|2PACz|CsFAMA-17|C$_{60}$|BCP|Al devices measured with simulated solar (AM1.5G, 100 mW cm$^{-2}$) illumination. The inset shows the external quantum efficiency (EQE) spectrum recorded with 1-sun equivalent bias illumination. **c** The sensitive EQE spectrum of the same solar cell, divided into three regions: the above-bandgap (gray), the band-edge (Urbach tail, red), and the sub-bandgap (yellow). The numbers 1 and 2 indicate two sub-bandgap defects. The dashed red line shows the blackbody radiation photon flux at 300 K (right axis). Source data are provided as a Source Data file.

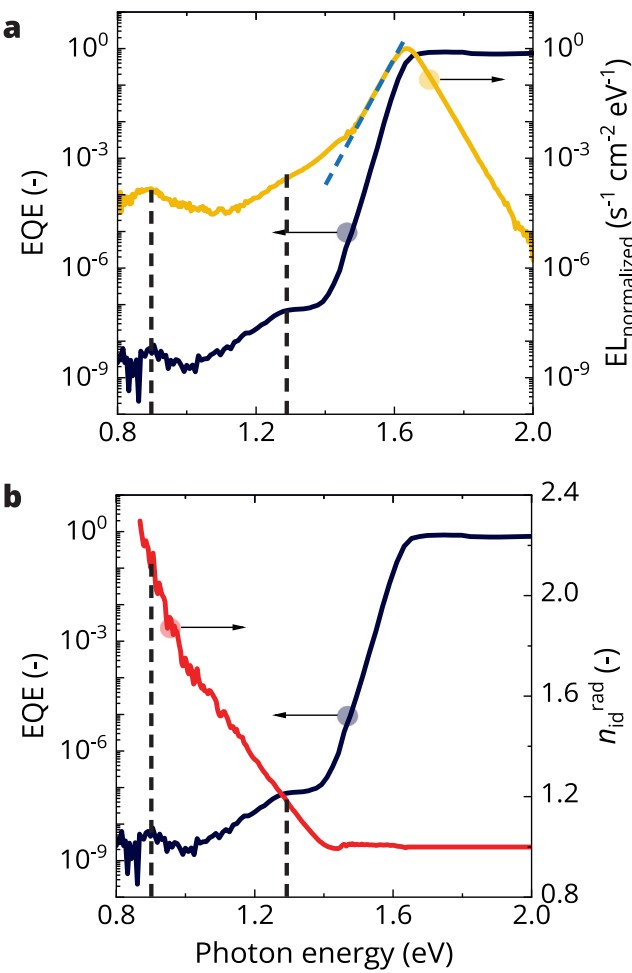

**Fig. 2 | Measured sensitive photocurrent and electroluminescence (EL) spectra of a CsFAMA-17 perovskite solar cell and the determined radiative ideality factor. a** External quantum efficiency (EQE) combined with a semilogarithmic plot of the normalized EL spectrum of the same device (yellow line), recorded at an internal voltage close to the open-circuit voltage ($V_{int} \sim V_{OC}$). **b** EQE combined with the radiative ideality factor ($n_{id}^{rad}$) (red line) calculated using Eq. (7). The black dashed lines in (**a**) and (**b**) are guides to the eye denoting the two defect energies. The blue dashed line in (**a**) shows the extrapolation of the band-to-band EL peak. The difference between the yellow line and the blue dashed line is attributed to EL from defects. Source data are provided as a Source Data file.

EL spectroscopy was used to measure the emission of CsFAMA-17 PSCs (Fig. 2a). The measurements (EQE and EL) were performed on the same solar cell to ensure identical optical interference effects[4,22,23]. The EL emission of the cell was stable under electrical bias for prolonged periods (Supplementary Fig. 3). At 1.63 eV, the EL emission corresponds to the BTB transition. Interestingly, below the bandgap two distinct sub-bandgap emissions are visible, centered at 0.89 and 1.28 eV, with much lower intensity (approx. five orders of magnitude lower) than the main emission peak. The energies of these emissions match the energies for sub-bandgap EQE contributions and therefore can be assigned to these sub-bandgap states.

According to the reciprocity principle[14], charge-generating states in the EQE should be emissive. In fact, from the product of EQE and the black-body radiation, the EL spectrum should readily follow according to:

$$\varphi_{em}(E) = \varphi_{bb}(E)Q_e^{PV}(E)\left\{\exp\left(\frac{qV_{int}}{n_{id}^{rad}k_BT}\right) - 1\right\} \quad (6)$$

where $\varphi_{em}(E)$ is the EL emission photon flux and $V_{int}$ is the internal voltage (see Supplementary Note 1)[12,24,25]. This relation describes the reciprocity of light emission and absorption of a solar cell based on the detailed balance, valid at thermal equilibrium, as derived by Rau[14]. Its validity further stems from the negligibility of non-linear effects and its applicability has been studied in depth by others in the field[23,26]. Note that in the original equation, the $n_{id}^{rad}$ is taken as 1[27]. However, similar to Müller et al.[28], we find that calculating the EL from EQE via Eq. (6) does not match the measured EL across the spectral range (Supplementary Fig. 4). In the sub-bandgap region, the EL calculated via Eq. (6) and the measured EL deviate strongly. Additionally, if we calculate the $V_{OC}^{rad}$ according to Eq. (2) using the sensitive EQE and Eqs. (3) and (4), we obtain $V_{OC}^{rad}$ = 1.15 V, which is very close to $V_{OC}$ = 1.10 V, and—erroneously—suggests very low non-radiative losses (vide infra).

Previously, a double-diode model, an extension of the Shockley diode model, has been proposed to explain the apparent breakdown of reciprocity[16]. The model assigns BTB recombination ($n_{id}$ = 1) kinetics to transitions involving above-bandgap states and SRH recombination ($n_{id}$ = 2) kinetics via deep defect states to the sub-bandgap transitions. However, sensitive measurements of EL and EQE, such as those presented here, allow for the calculation of the radiative ideality factor across the spectral range via the use of optical reciprocity. To calculate the radiative ideality factor for transitions at every photon energy ($n_{id}^{rad}(E)$), Eq. (7) is introduced (for a derivation see Supplementary

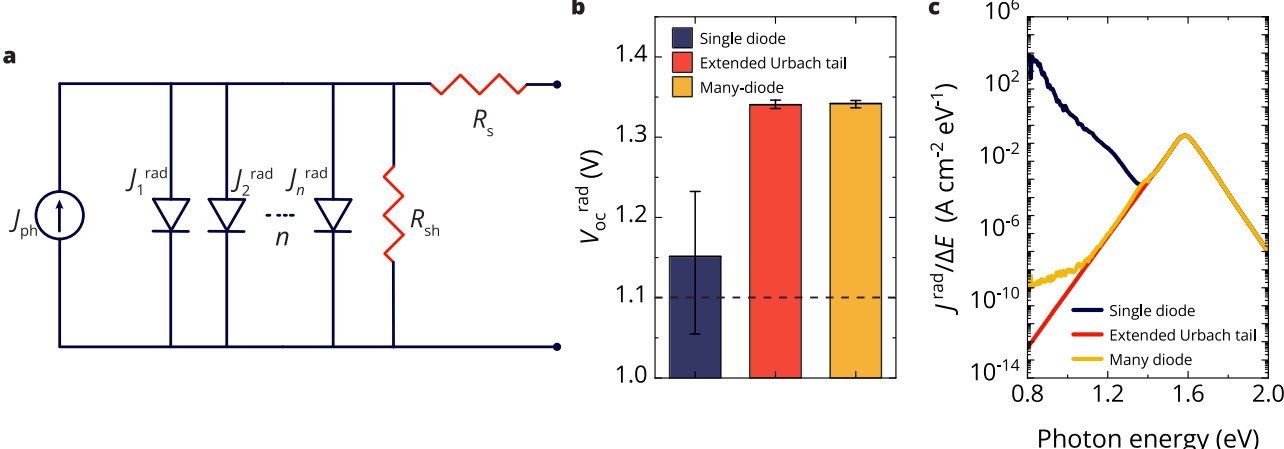

**Fig. 3 | The diode equation extended to a many-diode model incorporating an energy-dependent ideality factor. a** The equivalent circuit of the many-diode model. Included is the photocurrent ($J_{ph}$), the number of diodes ($n$) giving rise to radiative reverse saturation currents ($J_n^{rad}$), the series resistance ($R_s$), and the shunt resistance ($R_{sh}$). **b** Statistical distribution of the radiative open-circuit voltage ($V_{OC}^{rad}$) of CsFAMA-17 perovskite solar cells (PSCs) for a radiative ideality factor $n_{id}^{rad} = 1$ (dark blue), for $n_{id}^{rad} = 1$ with an extended Urbach tail (red), or for $n_{id}^{rad}(E)$ (yellow). The dashed line indicates the average experimentally measured open-circuit voltage ($V_{OC}$). Note the negative non-radiative voltage losses occurring for some of the single-diode calculations. Error bars indicate the standard deviation. **c** Reverse saturation current as function of energy ($J^{rad}(E)$) of a CsFAMA-17 PSC assuming $V_{OC}^{rad} = 1.34$ V for a radiative ideality factor $n_{id}^{rad} = 1$ (dark blue), for $n_{id}^{rad} = 1$ with an extended Urbach tail (red), or $n_{id}^{rad}(E)$ (yellow). The $J^{rad}(E)$ is divided by energy spacing ($\Delta E$) to enable integration. Source data are provided as a Source Data file.

Note 2) and is described as:

$$n_{id}^{rad}(E) = \left[ \frac{1}{n_{id,band}^{rad}} - \frac{k_B T}{q V_{int}} \ln \left( \frac{\varphi_{em}^{band} \varphi_{bb}(E) Q_e^{PV}(E)}{\varphi_{em}(E) \varphi_{bb}^{band} Q_{e,band}^{PV}} \right) \right]^{-1} \quad (7)$$

where $n_{id,band}^{rad}$ is the above-bandgap radiative ideality factor. The subscripts and superscripts "band" denote values taken at the energy of the emission maximum and therefore are independent of energy. Here, it is assumed that the transition at the emission maximum occurs in a BTB fashion with an ideality factor of 1 ($n_{id,band}^{rad} = 1$). As a result, based on this equation we can assess the radiative ideality factor, and thus the recombination kinetics, for transitions in the bandgap.

We note that $Q_e^{PV}(E)$ and $\varphi_{em}(E)$ must be measured on identical devices for Eqs. (6) and (7) to be valid. If these spectra are recorded on two separate devices, their respective sub-bandgap EQE signal can differ up to an order of magnitude due to differences in optical interference effects[4]. The effect of optical interference on the recorded spectra is mitigated in this work by recording EL and EQE on the same device, adhering to the prerequisites of the reciprocity theorem.

In Fig. 2b, the EQE spectrum and $n_{id}^{rad}(E)$ are plotted for a CsFAMA-17 PSC. All parameters were interpolated to a 3.4 meV energy spacing. The radiative ideality factor was set to unity above the bandgap and found to remain 1 along the Urbach tail. Interestingly, $n_{id}^{rad}$ rapidly increases when the defect contribution appears in the EQE spectrum. This rise occurs in a seemingly linear fashion reaching ~2 near mid-bandgap (~0.8 eV). This observation is markedly different from the step-like ideality factor assumed in the double-diode model which assumes all sub-bandgap states to behave in a SRH fashion for deep defect states. Instead, these observations imply that charge-carrier recombination kinetics depend on the nature of the defect states; as photon energy decreases and the defect becomes deeper, charge carriers recombine more akin to SRH kinetics for deep defect states. Strikingly, the method yields an ideality factor of ~2 for mid-bandgap transitions exactly as predicted by the SRH model. Note that variations in $T$ and $V_{int}$ (as a consequence of varied assumed series resistance ($R_s$)) only have a minor effect on the absolute value of $n_{id}^{rad}$, and have no influence on the observed relation with transition energy (Supplementary Fig. 5).

Conceptually, it is no surprise that defects with energy in between the bandgap and mid-bandgap transition facilitate recombination with an ideality factor between 1 and 2. We detail a mathematical approach to explain the observed behavior in Supplementary Note 3. This theoretical assessment is universal for band-to-band absorbers exhibiting defects in the bandgap and therefore is not limited to the perovskite semiconductors discussed here.

To incorporate an energy-dependent ideality factor of sub-bandgap states into the diode equation, we describe $J_0^{rad}$ as a summation of current contributions ($J_{\triangle E}^{rad}(E)$) similar to a Riemann sum. Here, $J_{\triangle E}^{rad}(E)$ is given as:

$$J_{\triangle E}^{rad}(E) = q Q_e^{PV}(E) \varphi_{bb}(E) \Delta E \quad (8)$$

where $\Delta E$ is the energy step size used in the interpolation of the measured EQE spectrum (3.4 meV here). The diode equation then takes the form of a summation of the photocurrent and the radiative saturation current contributions, amounting in principle to a many-diode model, described at open-circuit conditions as:

$$-J_{ph} + \sum_{i=E_1}^{E_2} J_{\triangle E}^{rad}(E_i) \left\{ \exp \left( \frac{q V_{OC}^{rad}}{n_{id}^{rad}(E_i) k_B T} \right) - 1 \right\} = 0 \quad (9)$$

where $E_1$ and $E_2$ are the energy boundaries of the experimental EQE spectrum. The equivalent circuit of the many-diode model is depicted in Fig. 3a, including the shunt and series resistance ($R_s$ and $R_{sh}$). Note that the many-diode equivalent circuit is not a practical approach to be used in device simulation. Using Eq. (9), $V_{OC}^{rad}$ can be calculated for an energy-dependent ideality factor in a numerical fashion. Additionally, the current contribution of the many diodes can be expressed by a single energy-dependent current density ($J^{rad}(E)$) yielding a simplified expression:

$$-J_{ph} + \sum_{i=E_1}^{E_2} J^{rad}(E_i) = 0 \quad (10)$$

We use Eq. (9) to calculate $V_{OC}^{rad}$ for a constant radiative ideality factor (single diode, $n_{id}^{rad} = 1$) or the calculated energy-dependent radiative ideality factor (many-diode, $n_{id}^{rad}(E)$) and show the results in Fig. 3b. Additionally, we include $V_{OC}^{rad}$ in the single-diode assumption for an EQE with an extended Urbach tail rather than the measured sub-

**Table 1 | Open-circuit voltage and voltage losses in CsFAMA-17, KCsFAMA-25, and KCsFAMA-40 p-i-n PSCs with/without choline chloride (CCl) passivation**

| | $E_g$ (eV) | $V_{OC}^{rad}$ (V) | $V_{OC}$ (V) | $\Delta V_{OC}^{non-rad}$ (mV) | $\Delta V_{OC}^{non-rad,film}$ (mV) | $\Delta V_{OC}^{non-rad,ETL}$ (mV) | $\Delta V_{OC}^{non-rad,HTL}$ (mV) |
|---|---|---|---|---|---|---|---|
| CsFAMA-17 | 1.63 | 1.342 | 1.104 | 238 | 139 ± 14 | 102 ± 20 | 8 ± 13 |
| KCsFAMA-25 | 1.68 | 1.382 | 1.125 | 257 | 189 ± 14 | 80 ± 15 | −2 ± 16 |
| KCsFAMA-40 | 1.78 | 1.475 | 1.162 | 313 | 170 ± 12 | 128 ± 14 | 21 ± 14 |
| CsFAMA-17 CCl | 1.63 | 1.342 | 1.150 | 199 | 152 ± 9 | 50 ± 14 | 2 ± 9 |
| KCsFAMA-25 CCl | 1.68 | 1.384 | 1.170 | 214 | 187 ± 13 | 36 ± 9 | −18 ± 15 |
| KCsFAMA-40 CCl | 1.78 | 1.485 | 1.240 | 245 | 172 ± 10 | 56 ± 16 | 14 ± 15 |

Included are the radiative open-circuit voltage ($V_{OC}^{rad}$) determined using the many-diode model, measured open-circuit voltage ($V_{OC}$), total non-radiative open-circuit voltage loss ($\Delta V_{OC}^{non-rad}$ (= $V_{OC}^{rad} - V_{OC}$)), and the non-radiative losses obtained by comparing quasi-Fermi level splitting (QFLS) and $V_{OC}^{rad}$ according to Eqs. (11) and (12). Standard deviations are included for the QFLS (see "Methods").

bandgap EQE. For the many-diode model, the solution is obtained numerically employing a 3.4 meV energy spacing. When the single-diode model is applied to several CsFAMA-17 PSCs, a mean $V_{OC}^{rad}$ of 1.15 V is obtained with a noticeable spread that originates from the varying noise floor in the highly sensitive EQE. This radiative voltage is just 50 mV larger than the experimentally measured $V_{OC}$ (indicated by the dashed line in Fig. 3b), which would mistakenly indicate extremely low non-radiative losses and significant radiative losses. For some cells, the $V_{OC}^{rad}$ is even lower than $V_{OC}$, similar to observations in the literature[16]. On the contrary, the many-diode model provides a mean $V_{OC}^{rad}$ of 1.34 V and $\Delta V_{OC}^{non-rad}$ of 240 mV. Note that purposefully ignoring the sub-bandgap EQE signals, by extending the Urbach tail, yields very similar results. The similarity between $V_{OC}^{rad}$ calculated by extending the Urbach tail and $V_{OC}^{rad}$ obtained from the many-diode model demonstrates that emissive sub-bandgap states do not affect $V_{OC}^{rad}$.

To visualize the effect of the radiative ideality factor on the contribution of the defect states, we plot in Fig. 3c the $J^{rad}(E)$ calculated for a constant radiative ideality factor (single-diode, $n_{id}^{rad} = 1$) and calculated for an energy-dependent radiative ideality factor (many-diode, $n_{id}^{rad}(E)$) following Eqs. (9) and (10). In the calculation, we take the $V_{OC}^{rad}$ (1.34 V) from the numerical solution of Eq. (9) with $n_{id}^{rad}(E)$. The single-diode model shows a strong increase in $J^{rad}(E)$ where the sub-bandgap EQE dominates ($E < 1.35$ eV), owing to the increasing black-body radiation with decreasing energy. For the many-diode model, however, the $J^{rad}(E)$ remains low in the sub-bandgap region compared to the above-gap region. Therefore, sub-bandgap states have a negligible effect on $\sum_{i=E_1}^{E_2} J^{rad}(E_i)$ for the many-diode model while having an outsized effect on $\sum_{i=E_1}^{E_2} J^{rad}(E_i)$ for the single-diode model. We note that a similar result could be obtained for the double-diode model where sub-bandgap states barely contribute to $\sum_{i=E_1}^{E_2} J^{rad}(E_i)$. The sub-bandgap defects contribute so little that assuming an exponentially decreasing Urbach tail rather than the measured sensitive EQE spectrum gives qualitatively equal $\sum_{i=E_1}^{E_2} J^{rad}(E_i)$, using the single-diode model, as displayed in Fig. 3c. Thereby, the many-diode model proves that the defect contribution in sensitive EQE can be safely omitted upon calculation of the $V_{OC}^{rad}$, and that halting the integration of the sensitive EQE in the detailed balance method at the onset of defect response allows to employ the simpler single-diode model.

From these measurements, we conclude that the radiative defects observed in EL and sensitive photocurrent spectroscopy have a radiative ideality factor scaling inversely with their transition energy, and subsequently have no effect on the $V_{OC}^{rad}$. As a consequence of the energy-dependent radiative ideality factor, the Shockley diode model needs to be extended to a many-diode model to describe the results and allow the calculation of $V_{OC}^{rad}$ taking into account sub-bandgap states. For straightforward $V_{OC}^{rad}$ determinations, the results of the many-diode model justify that neglecting the defect response is a valid method that circumvents the use of the more convoluted many-diode model.

## Non-radiative losses

Non-radiative recombination via sub-bandgap states could still have a significant influence on $\Delta V_{OC}^{non-rad}$ and subsequently on $V_{OC}$. However, given their non-radiative nature, these defect transitions are challenging to characterize. The impact of non-radiative recombination can be probed by determining the quasi-Fermi level splitting (QFLS) from the absolute photoluminescence spectrum[17,29–31]. The QFLS experiments can be performed on perovskite films or perovskite films with an adjacent electron transport layer (ETL) or hole transport layer (HTL), or both, and even on complete cells. By comparing with $V_{OC}^{rad}$, the non-radiative losses in the perovskite film can be established as originating from the perovskite film and the interfaces with the charge transport layers (CTL) via:

$$\Delta V_{OC}^{non-rad,film} = V_{OC}^{rad} - \frac{\Delta \mu^{film}}{q} \qquad (11)$$

$$\Delta V_{OC}^{non-rad,CTL} = \frac{\Delta \mu^{film} - \Delta \mu^{CTL}}{q} \qquad (12)$$

where $\Delta \mu$ is the QFLS. Note that the superscript "film" does not necessarily correspond to losses that are isotropic throughout the material, but rather denotes a film without transport layers. We note that the QFLS of a perovskite film can also deviate from that in a solar cell because the absorption of light by a perovskite film on glass may differ from (the same) perovskite film in a solar cell.

The absolute PL spectra of glass|CsFAMA-17, glass|ITO|2PACz|CsFAMA-17, and glass|CsFAMA-17|$C_{60}$ samples were measured (Supplementary Fig. 6), and their QFLSs were extracted and inserted into Eqs. (11) and (12) (Table 1). We find the perovskite film suffers from a -139 mV non-radiative loss. A negligible additional loss (-8 mV) arises from the HTL/perovskite (2PACz|CsFAMA-17) interface, while a significant extra loss (-102 mV) is induced by the perovskite/ETL interface (CsFAMA-17|$C_{60}$). The non-radiative recombination losses at the CsFAMA-17|$C_{60}$ interface are well described in the literature, but losses in the perovskite film are also substantial[31–33]. Note that the formation of perovskite on 2PACz can be different than on glass, which can lead to differences in the QFLS value obtained[34]. The different contributions are visualized in Fig. 4a depicting the $V_{OC}$, $V_{OC}^{rad}$, $\Delta V_{OC}^{non-rad}$, $\Delta V_{OC}^{non-rad,film}$, $\Delta V_{OC}^{non-rad,ETL}$, and $\Delta V_{OC}^{non-rad,HTL}$.

To test the universality of the many-diode model and its use in calculating and disentangling the radiative and non-radiative losses in PSCs, we performed experiments on n-i-p device architecture and p-i-n devices for perovskites with wider bandgaps obtained by increasing the bromide/iodide ratio, namely $K_{0.05}[Cs_{0.05}(FA_{0.75}MA_{0.25})_{0.95}]_{0.95}$ $Pb(I_{0.75}Br_{0.25})_3$ (KCsFAMA-25) and $K_{0.05}[Cs_{0.05}(FA_{0.60}MA_{0.40})_{0.95}]_{0.95}$ $Pb(I_{0.60}Br_{0.40})_3$ (KCsFAMA-40). For these higher bromide fractions, potassium is added to ensure stability against halide segregation under light and electrical bias (Supplementary Fig. 7) which otherwise might affect the defects present in the perovskite[35,36]. Additionally, choline chloride (CCl) passivation of the perovskite|$C_{60}$ interface and phenformin hydrochloride (PhenHCl) passivation of the perovskite bulk in

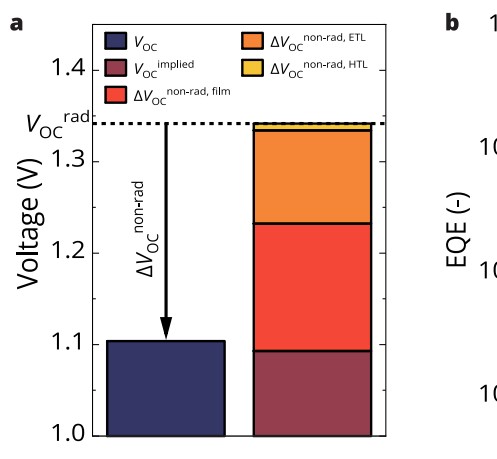 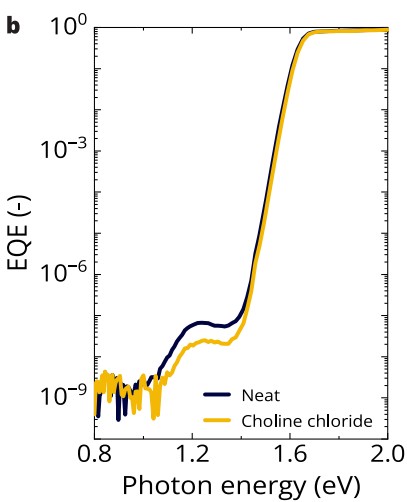 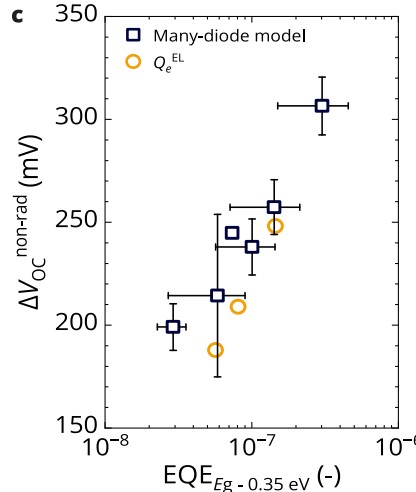

**Fig. 4 | Many-diode model applied to determine the non-radiative voltage losses in p-i-n CsFAMA-17 devices and correlated to external quantum efficiency (EQE) defect signal. a** Disentanglement of voltage losses in CsFAMA-17 p-i-n perovskite solar cells (PSCs). Comparison of the experimental open-circuit voltage ($V_{OC}$) determined from current-density voltage (J-V) sweeps and the non-radiative voltage losses of the pristine CsFAMA film ($\Delta V_{OC}^{non-rad,film}$), and CsFAMA film with electron and hole transporting layers ($\Delta V_{OC}^{non-rad,ETL}$, and $\Delta V_{OC}^{non-rad,HTL}$) determined from quasi-Fermi level splitting (QFLS) according to Eqs. (11) and (12), respectively. The implied open-circuit voltage ($V_{OC}^{implied}$) in the right bar is calculated from the $V_{OC}^{rad}$, $\Delta V_{OC}^{non-rad,film}$, $\Delta V_{OC}^{non-rad,ETL}$, and $\Delta V_{OC}^{non-rad,HTL}$ and slightly deviates from the measured $V_{OC}$ which we attribute to inaccuracies in the determination of the QFLS. The dashed line indicates the radiative open-circuit voltage ($V_{OC}^{rad}$) determined using the many-diode model. **b** Semilogarithmic plot of the sensitive EQE spectrum of a CsFAMA-17 solar cell without (dark blue) and with choline chloride (CCl) passivation (yellow). **c** Semilogarithmic plot of the average total non-radiative voltage losses ($\Delta V_{OC}^{non-rad}$) vs. the average EQE signal at energies 0.35 eV below the bandgap ($E = E_g - 0.35$ eV) for several p-i-n PSCs. From lowest to highest EQE signal, the perovskite compositions are CsFAMA-17 CCl, KCsFAMA-25 CCl, KCsFAMA-40 CCl, CsFAMA-17, KCsFAMA-25, and KCsFAMA-40. Error bars represent the standard deviations. As a reference, the $\Delta V_{OC}^{non-rad}$ obtained from the electroluminescence quantum efficiency ($Q_e^{EL}$) measurements is added for CsFAMA-17, KCsFAMA-25, and KCsFAMA-40 (listed lowest to highest EQE), showing good correspondence with $\Delta V_{OC}^{non-rad}$ values from the many-diode model. For the $Q_e^{EL}$ data, the horizontal axis value was determined as the EQE value of that specific cell and is in the range of the values for the other p-i-n PSCs. Source data are provided as a Source Data file.

p-i-n devices is studied (CsFAMA-17 CCl, KCsFAMA-25 CCl, KCsFAMA-40 CCl, and CsFAMA-17 PhenHCl). CCl and PhenHCl are known to improve the $V_{OC}$ and thus should affect either $V_{OC}^{rad}$, $\Delta V_{OC}^{non-rad}$, or both[37–39]. Also, the effect of aging under $N_2$-atmosphere on defect response and voltage losses of KCsFAMA-25 and KCsFAMA-40 p-i-n devices were studied.

Sensitive EQE and EL on an n-i-p PSC with a glass|ITO|SnO₂|PCBM| CsFAMA-17|PM6|MoO$_x$|Au device stack was measured and a similar defect contribution was recorded as for the p-i-n architecture (Supplementary Fig. 8a, b). Using these sensitive EQE and EL spectra, the many-diode model was applied to calculate $n_{id}^{rad}(E)$ (Supplementary Fig. 8c). A comparable radiative ideality factor was found for the n-i-p architecture, i.e., $n_{id}^{rad} = 1$ at the end of the Urbach tail and increasing to $n_{id}^{rad} \approx 2$ at mid-bandgap transitions (Supplementary Fig. 8d). The $V_{OC}^{rad}$ was determined to be 1.35 V, which is virtually identical to the value obtained for the p-i-n device (1.34 V). These results demonstrate the general applicability of the many-diode model and the consistency in the $n_{id}^{rad}(E)$ values derived from it.

The sensitive EQE and EL spectra of CsFAMA-17, KCsFAMA-25, KCsFAMA-40, and p-i-n PSCs without and with CCl passivation are shown in Supplementary Fig. 9. The $n_{id}^{rad}(E)$ derived from these spectra is depicted in Supplementary Fig. 10. We find—for all compositions—that above the bandgap and down to the bottom of the Urbach tail $n_{id}^{rad}(E) = 1$ but that $n_{id}^{rad}(E)$ increases for lower photon energies resulting in $n_{id}^{rad}(E) \approx 2$ at the mid-bandgap energy. For 3-months aged KCsFAMA-25 and KCsFAMA-40 p-i-n devices (without CCl), sensitive EQE and EL spectra (Supplementary Fig. 11) show an increase in sub-bandgap EQE signal upon aging. These spectra were used to determine $n_{id}^{rad}(E)$ (Supplementary Fig. 11), resulting in similar values regardless of the aging.

As expected, $V_{OC}^{rad}$ increases with increasing bandgap and bromide fraction, but $\Delta V_{OC}^{non-rad}$ also increases (Table 1), in agreement with the commonly observed voltage plateau for mixed-halide perovskites[40,41].

For the KCsFAMA-25 aged and KCsFAMA-40 aged devices, $V_{OC}^{rad}$ remains unchanged after 3 months while only for KCsFAMA-40 aged device the $\Delta V_{OC}^{non-rad}$ increases 90 mV (Supplementary Table 1). In QFLS experiments, we find a similar plateauing trend for KCsFAMA-25 and KCsFAMA-40 as for CsFAMA-17. The non-radiative losses stemming from the perovskite film and the perovskite|ETL interface are higher than from the HTL|perovskite interface, which gives a negligible loss (Table 1 and Supplementary Fig. 12). For KCsFAMA-25, the QFLS for glass|ITO|2PACz|KCsFAMA-25 is higher than for glass|KCsFAMA-25, amounting to a negative $\Delta V_{OC}^{non-rad,HTL}$. We tentatively ascribe this result to small changes in perovskite film formation on a glass substrate compared to the formation on a 2PACz-covered ITO substrate[34].

Adding PhenHCl to the CsFAMA-17 perovskite bulk results in a gain in $V_{OC}$ of 20 mV (Supplementary Fig. 13). Using sensitive EQE and EL, the many-diode model was used to determine $n_{id}^{rad}(E)$ for CsFAMA-17 PhenHCl PSCs and found to be similar to that for neat CsFAMA-17 devices without bulk passivation (Supplementary Fig. 14). Note that at lower photon energies, $n_{id}^{rad}(E)$ starts to deviate due to a higher $R_s$ determined for the CsFAMA-17 PhenHCl devices compared to CsFAMA-17. The $V_{OC}^{rad}$ of PhenHCl passivated device was determined to be 1.32 V, close to 1.34 V for the neat CsFAMA-17. From sensitive EQE and EL spectra, a decreased defect signal is found for CsFAMA-17 PhenHCl devices compared to the neat CsFAMA-17 devices (Supplementary Fig. 14a, b). The QFLS of partial device stacks showed that significant QFLS losses occur in the perovskite film when deposited on a glass substrate (Supplementary Fig. 14d). As mentioned before, we tentatively ascribe the low glass|perovskite QFLS to changes in the perovskite formation on glass compared to perovskite formation on 2PACz-covered ITO substrates[34], corroborated by the much higher QFLS obtained for glass|ITO|2PACz|perovskite. The strongly different perovskite formation on glass inhibits further deconvolution of non-radiative losses for PhenHCl passivated devices. Still, major QFLS

losses remain at the perovskite|fullerene interface, which seems not to be significantly improved by the PhenHCl bulk passivation.

Surface passivation of the perovskite films using CCl yields an increased $V_{OC}$ (Table 1). The many-diode model and the QFLS experiments show that CCl does not significantly affect $V_{OC}^{rad}$ but decreases $\Delta V_{OC}^{non-rad}$ (Table 1 and Supplementary Figs. 12 and 15). Predominantly, the $\Delta V_{OC}^{non-rad,ETL}$ was lowered significantly after CCl surface treatment, which is in accordance with the literature[31,37]. Examining the sensitive EQE spectra of PSCs without and with CCl passivation, we find identical EQE above the bandgap but a significantly decreased sub-bandgap EQE with CCl passivation (Fig. 4b and Supplementary Figs. 15c and 16). It is well-established that the EQE defect signal in a p-i-n configuration arises from a defect located near the perovskite|fullerene interface[4], and finding that CCl passivation reduces the defect-EQE signal is consistent with this result. Since, both $\Delta V_{OC}^{non-rad}$ (predominantly due to the $\Delta V_{OC}^{non-rad,ETL}$) and the defect-EQE signal are reduced by CCl passivation, the two seem to correlate.

To further test this correlation, we compare the defect-EQE signal measured for the different compositions (with/without CCl) to $\Delta V_{OC}^{non-rad}$. To circumvent difficulties with varying noise levels between samples, we use the EQE signal at 0.35 eV below $E_g$ as a gauge. Figure 4c shows $\Delta V_{OC}^{non-rad}$ as a function of the EQE at $E_g - 0.35$ eV on a semilogarithmic plot for cells of CsFAMA-17, KCsFAMA-25, and KCsFAMA-40, with and without CCl passivation. The $\Delta V_{OC}^{non-rad}$ obtained from electroluminescence quantum efficiency ($Q_e^{EL}$) measurements on a CsFAMA-17, KCsFAMA-25, and KCsFAMA-40 cell is added for comparison. A striking dependence of $\Delta V_{OC}^{non-rad}$ on the defect signal appears, where the voltage loss increases with increasing sub-bandgap EQE. Hence, in addition to identifying the location of such defects at the perovskite|$C_{60}$ interface, the defect signal in sensitive photocurrent measurements correlates with the $\Delta V_{OC}^{non-rad}$ losses in a p-i-n PSC. This finding opens avenues for further research into defect related non-radiative losses in mixed-halide lead perovskite solar cells using sensitive EQE. Comparing the $\Delta V_{OC}^{non-rad}$ derived using the many-diode model to the $\Delta V_{OC}^{non-rad}$ determined from the $Q_e^{EL}$ via $\Delta V_{OC}^{non-rad} = (-n_{id,1}^{rad} k_B T/q) \ln(Q_e^{EL})$, we find the values are in good mutual agreement (Fig. 4c). In this equation, $n_{id,1}^{rad}$ is the radiative ideality factor of the dominant emissive recombination pathway at $V_{OC}$ (and $V_{OC}^{rad}$). This equation is derived from the many-diode model in Supplementary Note 4, and is similar to results described in literature[27].

## Discussion

Sub-bandgap defect states probed using sensitive photocurrent and electroluminescence spectroscopy allow to assess their effect on the open-circuit voltage in CsFAMA-17 PSCs. These spectroscopic techniques identified the same low-energy defect states. We demonstrated that to maintain reciprocity of absorption and emission, the radiative ideality factor ($n_{id}^{rad}(E)$) should vary for defect states positioned across the bandgap. The $n_{id}^{rad}(E)$ is 1 above the bandgap and in the Urbach tail, but increases to ~2 for defect states located in the middle of the bandgap. To incorporate a photon-energy-dependent ideality factor in the mathematical description of a solar cell, the Shockley diode model was extended to a many-diode model. Following the many-diode model, radiative defects recorded in sensitive EQE and EL measurements do not affect radiative voltage losses. Applying the many-diode model thus allows for accurately determining $V_{OC}^{rad}$ from these spectral data and justifies that neglecting the defect response is a valid method to straightforwardly determine $V_{OC}^{rad}$.

To further unravel the contributions of sub-bandgap defect states to the voltage losses, the $V_{OC}^{rad}$ derived from the many-diode model can be combined with QFLS experiments. From this comparison, we conclude that in PSCs with a 2PACz|CsFAMA-17|$C_{60}$ configuration, the non-radiative losses mainly occur in the CsFAMA-17 PSCs perovskite film and at the CsFAMA-17 PSCs|$C_{60}$ interface.

The procedure of measuring sensitive EQE, EL, and non-radiative losses through QFLS was extended to an n-i-p architecture and to wide-bandgap (higher bromide fraction) perovskites, with and without CCl surface passivation, to test the generality of the results. Indeed, the same behavior was found for the $n_{id}^{rad}(E)$ for all PSCs resulting in a negligible influence of sub-bandgap defects on $V_{OC}^{rad}$. Upon CCl passivation, the non-radiative losses originating from the perovskite|$C_{60}$ interface are significantly reduced, but they remain the dominant contribution to the $\Delta V_{OC}^{non-rad}$. By comparing $\Delta V_{OC}^{non-rad}$ and the defect-EQE signal intensity for the different perovskites, a clear correlation was found. The $\Delta V_{OC}^{non-rad}$ determined from the many-diode model is also in accordance with the experimental data from $Q_e^{EL}$ measurements.

The many-diode model developed in this work shows that radiative sub-bandgap states do not affect the $V_{OC}^{rad}$ and justifies the common use of the single-diode model and a spectral cutoff to determine the $V_{OC}^{rad}$. Photocurrent spectroscopy signals of radiative sub-bandgap defects show a correlation to non-radiative recombination losses, but do not affect $V_{OC}^{rad}$ when the radiative ideality factor is taken into account.

## Methods

### Materials

Lead iodide ($PbI_2$, 99.99% trace metal basis) and lead bromide ($PbBr_2$, >98%) were purchased from TCI Chemicals. Methylammonium bromide (MABr) and formamidinium iodide (FAI) were purchased from Greatcell Solar Materials. Potassium iodide (KI, 99.999%) and cesium iodide (CsI, 99.999%) were purchased from Fisher Scientific and Sigma-Aldrich, respectively. [2-(9H-carbazol-9-yl)ethyl] phosphonic acid (2PACz, >98.0%) was purchased from TCI Chemicals. [6,6]-phenyl-$C_{61}$-butyric acid methyl ester (PCBM, 99%) was purchased from Solenne BV. PM6 was purchased from Solarmer Materials. Tin(IV) oxide ($SnO_2$ nanoparticles, 15 wt% in water) was purchased from Alfa Aesar. Dimethylformamide (DMF, 99.8%), dimethyl sulfoxide (DMSO, anhydrous 99.9%), anisole (99.7%), propan-2-ol (IPA, 99.95%), and chlorobenzene (CB, 99.8%) were purchased from Sigma-Aldrich. Ethanol (>95%) was purchased from Acros Organics.

### Solution preparation

Triple-cation mixed-halide $Cs_{0.05}(FA_{1-x}MA_x)_{0.95}Pb(I_{1-x}Br_x)_3$ perovskite solutions were prepared by mixing $FAPbI_3$ and $MAPbBr_3$ (1.5 mol l$^{-1}$) in DMF:DMSO 4:1 (v/v) solutions. First, $FAPbI_3$ solution (1.5 mol l$^{-1}$) was prepared by dissolving FAI (207.2 mg, 1.20 mmol) and $PbI_2$ (671.7 mg, 1.46 mmol) in DMF (776.9 µl) and DMSO (194.1 µl). $MAPbBr_3$ solution (1.5 mol l$^{-1}$) was prepared by dissolving MABr (28.2 mg, 0.252 mmol) and $PbBr_2$ (108.7 mg, 0.296 mol) in DMF (159.0 µl) and DMSO (40.0 µl). Both $FAPbI_3$ and $MAPbBr_3$ solutions were allowed to fully dissolve under stirring at 60 °C for 15 min. Thereafter, $FAPbI_3$ and $MAPbBr_3$ are mixed in the appropriate ratio to yield the desired Br:I ratio. Then, CsI (1.5 mol l$^{-1}$ in DMSO) is added to the FAMA solution yielding CsFAMA solution $[Cs_{0.05}(FA_{1-x}MA_x)_{0.95}Pb(I_{1-x}Br_x)_3]$. Furthermore, for KCsFAMA-25 and KCsFAMA-40 solutions, an additional KI solution (1.5 mol l$^{-1}$ in 4:1 DMF:DMSO) is added to the CsFAMA mixture. This results in $K_{0.05}[Cs_{0.05}(FA_{1-x}MA_x)_{0.95}]_{0.95}Pb(I_{1-x}Br_x)_3$ perovskite solutions ready for spin coating. HTL solutions were prepared by dissolving 2PACz in ethanol (0.33 mg ml$^{-1}$) and sonicating for 30 min before spin coating. Choline chloride (CCl) was dissolved in IPA (1 mg ml$^{-1}$) with overnight stirring at 60 °C. Tin(IV) oxide nanoparticle solutions were prepared by diluting (1:5 v:v) aqueous 15 wt% $SnO_2$ colloidal with water and stirring overnight at room temperature. PCBM was dissolved in CB (10 mg ml$^{-1}$). PM6 was dissolved in CF (5 mg ml$^{-1}$) with stirring at 60 °C.

## Device fabrication

Pre-patterned indium tin oxide (ITO) glass substrates (Naranjo Substrates) were cleaned by sonication in acetone for 15 min, followed by 30 s scrubbing and subsequent sonication with an aqueous dodecyl sodium sulfate (Acros, 99%) solution for 15 min. Thereafter, the substrates were rinsed using deionized water for 10 min after which they were sonicated in IPA for 15 min. The substrates were dried with a nitrogen gun, treated with ultraviolet (UV)-ozone for 30 min, and transferred to a nitrogen-filled glovebox. For devices, 2PACz solution (120 µl) was spin-coated for 30 s at 3000 rpm followed by annealing for 10 min at 100 °C. For n-i-p devices, $SnO_2$ solution (150 µl) was spin-coated for 60 s at 2800 rpm (acceleration of 2000 rpm s$^{-1}$) and the substrate was annealed at 150 °C for 30 min. Then, the PCBM solution (100 µl) was spin-coated atop for 60 s at 1000 rpm and the film was annealed for 10 min at 100 °C. All perovskite films were fabricated by spin coating the respective solutions for 35 s at 4000 rpm (acceleration of 800 rpm s$^{-1}$). During spinning (after 25 s), the substrate was washed with 300 µl anisole (antisolvent). The substrate was annealed at 100 °C for 30 min. For devices passivated with CCl, the CCl solution was deposited on annealed perovskite films spinning at 4000 rpm and thereafter annealed at 100 °C for 30 min. For the ETL of p-i-n devices, 20 nm fullerene $C_{60}$ followed by 8 nm bathocuproine (BCP) were thermally evaporated under a high vacuum followed by 100 nm of aluminum as an electrode. For n-i-p devices, PM6 solution (100 µl) was spin-coated dynamically atop the perovskite layer for 30 s at 3000 rpm, followed by annealing for 10 min at 110 °C. Then, $MoO_x$ (10 nm) and gold (100 nm) as electrodes were thermally evaporated under a high vacuum. The overlap between the bottom ITO and top metal electrodes yielded solar cell active areas of 0.09 or 0.16 cm$^2$.

## Current density–voltage characteristics

The cells were tested in a N$_2$-filled glove box at ambient temperature. To emulate ~100 mW cm$^{-2}$ AM1.5G light, a tungsten halogen lamp in combination with a Schott GG385 UV filter and Hoya LB120 daylight filter were used. Incident light was referenced using a Si photodiode. Shadow masks of 0.0676 or 0.1296 cm$^2$ were used to define the illuminated area of the solar cell. J-V characteristics were determined using a Keithley 2400 SMU. The J-V scan swept the applied voltage bias (with no pre-biasing) from +1.5 to −0.5 V for a reverse scan, or from −0.5 to +1.5 V for a forward scan using a scan rate of 0.25 V s$^{-1}$.

## External quantum efficiency

For regular EQE measurements, a tungsten halogen lamp (Philips Focusline, 50 W) was used and its light was mechanically chopped at 165 Hz (Stanford Research SR540) before passing through a monochromator (Oriel Cornerstone 130) and an aperture (0.0314 cm$^2$). The cell response was measured using a low-noise current pre-amplifier (Stanford Research SR570) in combination with a lock-in amplifier (Stanford Research SR830). The incident light intensity was referenced using a Si detector. 1-Sun light bias was simulated using a 530 nm LED (Thorlabs M530L3) driven by a Thorlabs DC4104 driver to accurately determine $J_{sc}$ under approximately AM1.5G conditions. Highly sensitive EQE measurements used the light from an Osram 64655 HLX 250 W tungsten halogen lamp mechanically chopped at 333 Hz passing appropriate sorting filters and dispersed using an Oriel Cornerstone 260 monochromator. The response was recorded using a Stanford Research SR570 pre-amplifier and a Stanford Research SR830 lock-in amplifier. Calibration was performed using reference Si and InGaAs detectors. The measured highly sensitive EQE spectra were scaled to regular EQE data.

## Electroluminescence

Devices were mounted in a home-built sample holder that maintained an inert atmosphere. A voltage bias of 1.5 or 1.7 V was applied to the PSC using a Keithley 2601 SMU. Electroluminescence spectra were recorded by an Edinburgh Instruments FLSP920 double-monochromator spectrometer with a near-infrared (NIR)-sensitive nitrogen-cooled (−80 °C) Hamamatsu R5509-73 photomultiplier and a UV-vis-sensitive multi-alkali Hamamatsu photomultiplier tube. The emission spectrum in the NIR region was measured after passing an 830 or 850 nm long-pass filter. Both UV-vis (above-bandgap) and NIR (sub-bandgap) spectra were corrected for the detector sensitivity. The UV-vis and NIR spectra could be unified, yielding a spectral range of 600 to 1600 nm, since they were measured consecutively without perturbing the setup in any way. To convert the EL spectrum to an energy scale, the spectra were subjected to a Jacobian transformation[42].

## Electroluminescence external quantum efficiency

To determine the external quantum efficiency of EL ($Q_e^{EL}$), devices were mounted atop a Si photodiode with a known photocurrent response. A bias was applied (Keithey 2400 source measure unit) to the PSC and the current was allowed to equilibrate after which the photocurrent generated by the silicon photodiode was measured (Keithey 2400 source measure unit). The applied bias was cycled from 0.8 V (below $V_{OC}$) to 1.6 V (above $V_{OC}$). The $Q_e^{EL}$ was determined as a function of injected current by dividing the output current of the silicon diode by the injected current, taking into account the photodiode response. The $Q_e^{EL}$ displayed in Fig. 4c corresponds to injection currents similar to $J_{SC}$.

## Radiative ideality factor

The radiative ideality factor at every photon energy ($n_{id}^{rad}(E)$) was determined from Eq. (7) using a home-built MATLAB script (Supplementary Code 1).

## Quasi-Fermi level splitting

QFLS was assessed through absolute photoluminescence (APL) measurements where excitation was achieved using a 455 nm Thorlabs M455F3 fiber-coupled LED. Samples were placed in an Avantes AvaSphere-30-REFL integrating sphere equipped with in-line filter holders for excitation light and emitted light, holding a 550 nm short-pass filter (Edmund Optics) and a 550 nm long-pass filter (Edmund Optics), respectively. The incident photon flux was adjusted to simulate AM1.5G conditions. The integrating sphere was connected to an Avantes AvaSpec-HSC1024X58TEC-EVO spectrometer by an optical fiber. The setup was calibrated using an Avantes halogen lamp yielding a spectral correction factor. Spectral photon fluxes ($\varphi_{PL}$) were obtained after a Jacobian transformation. Using a nonlinear least squares fit method in MATLAB, the QFLS was determined from the $\varphi_{PL}$. The relation between QFLS and photon flux is defined as follows:

$$\varphi_{PL}(E) = \frac{1}{4\pi^2\hbar^3 c^2} \frac{a(E)E^2}{\exp\left(\frac{E-\Delta\mu}{k_B T}\right) - 1} \quad (13)$$

where $a(E)$ is the photon energy-dependent absorptivity[43,44], it is assumed to be unity for photon energies sufficiently larger than the optical bandgap. Each film or (partial) stack combination was measured at least 3 times, the found QFLS value was averaged and the standard deviation was determined.

## Reporting summary

Further information on research design is available in the Nature Portfolio Reporting Summary linked to this article.

## Data availability
Source data are provided with this paper.

## Code availability
The MATLAB code to calculate the radiative ideality factor is provided with this paper.

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

## Acknowledgements

The authors are grateful for fruitful discussions with Thomas Kirchartz (Forschungszentrum Jülich, Germany) regarding the radiative ideality factor. We acknowledge funding from the Netherlands Ministry of Education, Culture, and Science (Gravity program 024.001.035) (T.P.A.v.d.P., R.A.J.J.) and the Netherlands Organization for Scientific Research (NWO) for funding through the Joint Solar Programme III (project 680.91.011) (K.D., R.A.J.J.) and the Spinoza prize (G.J.W.A., W.H.M.R., M.M.W., R.A.J.J.).

## Author contributions

G.J.W.A. and K.D. fabricated the samples. G.J.W.A. analyzed the devices and performed the calculations. G.J.W.A. and T.P.A.v.d.P. developed the underlying theory. W.H.M.R. carried out absolute PL measurements. G.J.W.A. and T.P.A.v.d.P. wrote the manuscript with help from K.D. and input from all co-authors. T.P.A.v.d.P. and K.D. devised the project. M.M.W. and R.A.J.J. supervised the project.

## Competing interests

The authors declare no competing interests.
