## [Peer Review File · Nature Communications]

Effect of sub-bandgap defects on radiative and non-radiative open-circuit voltage losses in perovskite solar cellsREVIEWER COMMENTS

Reviewer #1 (Remarks to the Author):

This work is of good quality and illustrates a more accurate way of estimating the non-radiative open circuit voltage loss in perovskite solar cells. Moreover, the paper describes that the non-ideality factor varies linearly in the sub-bandgap and reaches a value of 2 close to the mid-gap position, which is an interesting finding and has not been reported before. This study meticulously entailed the methodology used and all the results including sensitive photocurrent measurement, Electroluminescence, and photoluminescence characterization. However, there are some issues that are highlighted below and should be addressed. I believe incorporating these changes will further improve the quality of this manuscript. Therefore, I recommend publishing this paper after the suggested changes are made.

1. Line 36 of Introduction: Some discussion should be included in the introduction to elucidate the previous studies that have explored the non-radiative recombination losses in perovskite solar cells using other techniques, such as electrical (capacitance frequency) etc.
2. The significance of the use of many-diode model should be emphasized in the introduction and the drawbacks of the single and double-diode models should be highlighted.
3. The methodology developed to calculate the non-radiative Voc loss is used to study the effect of CCl passivation at the perovskite/ETL interface. However, the study can be enhanced by illustrating the effect of additives as defect passivation in the perovskite film, thereby finding the reduction in non-radiative Voc loss on incorporating the additives. This will prove to be more relevant for the readership as a vast majority of researchers are also trying to address the defects in the perovskite films.
4. Also, can this study be used to quantify the degradation of the perovskite solar cells when exposed to moisture, light and heat? Perhaps, non-radiative losses and their variation with degradation time can be a measure of the operation stability of the perovskite cells. The use of this methodology to compare the stability of cells without CCl passivation with the cells with passivation would demonstrate this application.

Reviewer #2 (Remarks to the Author):

In the work by Janssen and co-workers, the quantification and effect of radiative and nonradiative losses induced by sub-bandgap defect states in perovskite solar cells are investigated. Understanding the role of sub-bandgap defects in perovskite solar cells is crucial and while being a hot topic for many years, their influence on radiative and nonradiative voltage losses has remained unclear. Here, the authors employ very sensitive photovoltaic external quantum efficiency and electroluminescence measurements to probe the spectral sub-bandgap absorption and emission in the solar cells. Utilizing the reciprocity between absorption and emission and the concept of the radiative ideality factor, they devise a method based on a many-diode model to extract the contribution from sub-bandgap states to the overall radiative recombination loss. Building upon this method, they are further able to estimate the separate contributions from different radiative and nonradiative voltage loss channels for different perovskite materials and solar cell devices.

The paper is very well written, and the results are clearly and convincingly presented. Furthermore, the results and conclusions of this paper are of high impact not only to the perovskite research field but also to the broader semiconductor and photovoltaics communities. I am therefore willing to support publication of this paper in Nature Communications, provided that the authors address the following comments:

- Regarding the radiative ideality factor ($n_{id,rad}$) in Fig. 2b and more generally in the determination of $n_{id,rad}$: especially within the deep sub-bandgap region, how big is the influence of optical interference effects? Does this skew the value of the radiative ideality factor? Or does it matter? Please elaborate on this further in the main paper.

- The yellow circles in Fig 4c are the nonradiative V_{oc} loss as estimated from the electroluminescence quantum efficiency ($Q_{e,EL}$) using $dV_{oc,nr} = (-kT/q) \ln[Q_{e,EL}]$, as stated at the end of page 19 (note there seems to be a missing minus sign). However, if I understood things correctly, this relation is only valid for $n_{id,rad} = 1$; in other cases it should be replaced by $dV_{oc,nr} = (-n_{id,rad} * kT/q) \ln[Q_{e,EL}]$ (see your Ref 24).

- I would appreciate it if the authors could comment on the validity of Eq. (6). I presume this assumes some type of quasi-equilibrium condition. What does this mean for perovskite solar cells with substantial hysteresis where possible time dependences may be involved?

- Finally, the authors mention on page 8 and 9 in main text that an ideality factor of 2 is expected in case of SRH kinetics. However, this is not strictly true as it depends on trap depth and quasi-Fermi level, as discussed in the Supporting Information, but also defect distribution. In general, SRH via band-tail states

is expected to show an ideality factor less than 2, while an ideality factor of 2 is expected for states close to the middle of the gap. Therefore, on page 8 and 9, it is more accurate to say “SRH kinetics via deep defect states” rather than just SRH.

Reviewer #3 (Remarks to the Author):

In this communication article, Janssen et al. report a sensitive external quantum efficiency (EQE) and electroluminescence (EL) spectroscopy technique to assess radiative and non-radiative recombination losses in wide-bandgap mixed-halide lead perovskite solar cells. The authors propose a photon energy-dependent many-diode model to analyze the radiative recombination losses associated with subgap shallow defects and combine the photoluminescence quantum yield (PLQY) measurement of the quasi-Fermi level splitting limited by bulk and interface non-radiative recombination. Finally, the authors utilize devices with different perovskite compositions and choline chloride (CCI) passivation to validate the methodology.

This is overall a well-organized study with precise electrical and optical measurements and detailed modeling. However, the method proposed in this study and its conclusions are significantly important to photovoltaic research. The probe and analysis of radiative and non-radiative losses in a solar cell using EL/PLQY spectroscopy are well-established methods in the field of PV research. What the authors propose is not a sufficiently new concept or a new approach for publication in Nature Communications.

The paper needs more detailed discussions on how the developed method outperforms what the field knows. There is no need to use such a complex model to differentiate radiative and non-radiative recombination. The current techniques (EQE, PLQY, and ELQY) to probe radiative and nonradiative recombination are well-known and used by the field.

One main argument that conventional photocurrent spectroscopy measurements often overestimate the radiative open-circuit voltage (V_{oc}) loss, yielding unphysically low values of V_{oc} , is debatable. The key is that black-body radiation (BBR) from a semiconductor (Eqs. 4) cannot extend unlimitedly to zero photon energy. The shallow-defect assisted subgap radiative recombination cannot be treated simply as a band-to-band transition and contributes directly to the radiative recombination current by extending BBR integration.

The so-called single-diode model predicted V_{oc-rad} of ~ 1.15 V (Fig. 3b) could be simply estimated from the V_{oc-rad} of an ideal semiconductor with a bandgap of ~ 1.43 eV, which corresponds to the band tail of the measured real cell (Fig. 2). The V_{oc-rad} values shown in Fig. 3b and listed in Table 1 can be obtained through a simple and ideal semiconductor detail balance calculation based on corresponding bandgap values (i.e., EL peak value). The impact of subgap emission on V_{oc-rad} is negligible if a simple BBR with band cutoff is used. There is no need to run this many-diode model.

On the model side, the many-diode model shown in Fig. 2a is not a physical and practical equivalent circuit model when simulating the complete J-V characteristics.

First, photon energy should not be defined as discrete segments based on the photocurrent (EQE) measurement. It does not make any physical sense when considering the J-V relationship in a diode with a characteristic photon energy. Also, numerical interpolation should be used in the integration to solve the equation rather than a simple summation.

Furthermore, the two-diode model for radiative and non-radiative diodes is sufficient for practical device simulation. It is impractical to simulate the complete device performance for many but finite diodes. The authors can try it and will find out it is unrealistic for many diodes with R_s and R_{sh} .

Lastly, there are insufficient new findings and significant results about the wide-bandgap mixed-halide lead perovskite solar cells.

RESPONSE TO REVIEWERS COMMENTS

Reviewer #1:

This work is of good quality and illustrates a more accurate way of estimating the non-radiative open circuit voltage loss in perovskite solar cells. Moreover, the paper describes that the non-ideality factor varies linearly in the sub-bandgap and reaches a value of 2 close to the mid-gap position, which is an interesting finding and has not been reported before. This study meticulously entailed the methodology used and all the results including sensitive photocurrent measurement, Electroluminescence, and photoluminescence characterization. However, there are some issues that are highlighted below and should be addressed. I believe incorporating these changes will further improve the quality of this manuscript. Therefore, I recommend publishing this paper after the suggested changes are made.

We are thankful to the reviewer for the favorable comments on our work and address the concerns below.

1. Line 36 of Introduction: Some discussion should be included in the introduction to elucidate the previous studies that have explored the non-radiative recombination losses in perovskite solar cells using other techniques, such as electrical (capacitance frequency) etc.

We agree with the reviewer that non-optical techniques to probe defect states in thin semiconductor films are a useful addition compared to optical techniques. Therefore, we have now added a discussion on frequency-domain characterization of defect states to the main text on page 2.

2. The significance of the use of many-diode model should be emphasized in the introduction and the drawbacks of the single and double-diode models should be highlighted.

We have added additional emphasis on drawbacks of the single- and double-diode models, and highlighted the significance of the many-diode model developed in the introduction and conclusion (page 4 and 23).

3. The methodology developed to calculate the non-radiative Voc loss is used to study the effect of CCl passivation at the perovskite/ETL interface. However, the study can be enhanced by illustrating the effect of additives as defect passivation in the perovskite film, thereby finding the reduction in non-radiative Voc loss on incorporating the additives. This will prove to be more relevant for the readership as a vast majority of researchers are also trying to address the defects in the perovskite films.

We acknowledge the relevance of perovskite film passivation and have extended our voltage loss analysis to CsFAMA-17 devices that use phenformin hydrochloride (PhenHCl) as bulk additive. The results show that sub-bandgap defect contributions in sensitive EQE and EL are reduced when using PhenHCl, while the V_{OC} improved. The radiative ideality factors $n_{id}^{rad}(E)$ are similar between PhenHCl passivated and neat devices showing that recombination kinetics are the same. We commented on this on page 17, 18, and 19 and included additional figures (Supplementary Fig. 13 and 14) to corroborate this.

4. Also, can this study be used to quantify the degradation of the perovskite solar cells when exposed to moisture, light and heat? Perhaps, non-radiative losses and their variation with degradation time can be a measure of the operation stability of the perovskite cells. The use of this methodology to compare the stability of cells without CCl passivation with the cells with passivation would demonstrate this application.

Understanding how defect states and recombination kinetics affect degradation processes in perovskite solar cell is of utmost importance for long-term stability. Our method can be employed on degraded, segregated, or alike devices and a voltage loss analysis can be performed giving insight in changes in V_{oc}^{rad} , $\Delta V_{oc}^{non-rad}$, and QFLS. We have now added a loss analysis on three months aged (N_2 -stored) KCsFAMA-25 and KCsFAMA-40 p-i-n devices and commented on this on page 17 and 18 and included an additional figure and table (Supplementary Fig. 11 and Supplementary Table 1).

Reviewer #2:

In the work by Janssen and co-workers, the quantification and effect of radiative and nonradiative losses induced by sub-bandgap defect states in perovskite solar cells are investigated. Understanding the role of sub-bandgap defects in perovskite solar cells is crucial and while being a hot topic for many years, their influence on radiative and nonradiative voltage losses has remained unclear. Here, the authors employ very sensitive photovoltaic external quantum efficiency and electroluminescence measurements to probe the spectral sub-bandgap absorption and emission in the solar cells. Utilizing the reciprocity between absorption and emission and the concept of the radiative ideality factor, they devise a method based on a many-diode model to extract the contribution from sub-bandgap states to the overall radiative recombination loss. Building upon this method, they are further able to estimate the separate contributions from different radiative and nonradiative voltage loss channels for different perovskite materials and solar cell devices.

The paper is very well written, and the results are clearly and convincingly presented. Furthermore, the results and conclusions of this paper are of high impact not only to the perovskite research field but also to the broader semiconductor and photovoltaics communities. I am therefore willing to support publication of this paper in Nature Communications, provided that the authors address the following comments:

We thank the reviewer for carefully reading our manuscript and we are very grateful for the valuable comments.

- Regarding the radiative ideality factor ($n_{id,rad}$) in Fig. 2b and more generally in the determination of $n_{id,rad}$: especially within the deep sub=bandgap region, how big is the influence of optical interference effects? Does this skew the value of the radiative ideality factor? Or does it matter? Please elaborate on this further in the main paper.

Optical interference effects have been shown to change the sub-bandgap EQE spectrum up to an order of magnitude (*Nat Commun* **13**, 349 (2022)). However, optical interference influences emission and absorption equally and therefore does not play a role in the reciprocity of a solar cell. In fact, one

prerequisite to the validity of equation (6) is identical optical interference effects in EQE and EL. In our work, we record EQE and EL on the same solar cell to ensure that optical interference effects are indeed identical for both measurements. We have elaborated on this in the main text on page 9.

We illustrate the effect on n_{id}^{rad} of erroneously using EL and EQE of two different cells by a simple example:

Assume a φ_{em} measured on cell 1 and Q_e^{PV} measured on cell 2 used to calculate a single n_{id}^{rad} . If Q_e^{PV} is 10 times as large for cell 2 compared to cell 1, which is a realistic value for interference effects near contacts, Equation (6) readily shows that this translates to a substantial $\Delta n_{id}^{rad} = \ln(10) \times qV_{int}/(k_B T)$.

- The yellow circles in Fig 4c are the nonradiative Voc loss as estimated from the electroluminescence quantum efficiency ($Q_{e,EL}$) using $dV_{oc,nr} = (-kT/q) \ln[Q_{e,EL}]$, as stated at the end of page 19 (note there seems to be a missing minus sign). However, if I understood things correctly, this relation is only valid for $n_{id,rad} = 1$; in other cases it should be replaced by $dV_{oc,nr} = (-n_{id,rad} * kT/q) \ln[Q_{e,EL}]$ (see your Ref 24).

This comment led us to derive the equation brought up by the reviewer (originating from Ref 27, *Phys. Rev. Appl.* **7**, 044016 (2017)) for the many-diode model. The derivation is described in Supplementary Note 4 and yields a similar equation as found in *Phys. Rev. Appl.* **7**, 044016 (2017). We find that the $n_{id,1}^{rad}$ in this equation corresponds to the dominant emissive recombination at V_{oc} , which has a radiative ideality factor of 1 in our case.

We have corrected the equation on page 21 and refer to its derivation in Supplementary Note 4 (page 22-24). We also included the missing minus sign which the reviewer pointed out.

- I would appreciate it if the authors could comment on the validity of Eq. (6). I presume this assumes some type of quasi-equilibrium condition. What does this mean for perovskite solar cells with substantial hysteresis where possible time dependences may be involved?

Equation (6) is based on the reciprocity theorem and is adapted from U. Rau (*Phys. Rev. B* **76**, 085303 (2007)) by including a variable radiative ideality factor. This equation stems from the detailed balance, which is valid at thermal equilibrium. The situation at thermal equilibrium is extrapolated to non-equilibrium situations by using Shockley's diode equation, thereby assuming negligible non-linear contributions. There is a large body of work exploring the limits of the reciprocity theorem for different solar cells like (*Phys. Stat. Sol. (a)* **205**, 2737-2751 (2008)) and (*Phys. Rev. Appl.* **5**, 054003 (2016)).

We now explain the underlying assumptions and point towards studies on the limitations of the reciprocity theorem in the main text on page 8.

Hysteresis in perovskite solar cells will change the solar cell in time, also influencing the EQE and EL spectrum. The hysteresis does not affect the validity of the reciprocity theorem at any singular point in time, but spectra recorded at different times cannot be compared as the solar cell will have evolved due to the hysteresis. In that case, effectively you are not measuring/comparing the same solar cell anymore, which is a prerequisite for the reciprocity theorem.

- Finally, the authors mention on page 8 and 9 in main text that an ideality factor of 2 is expected in case of SRH kinetics. However, this is not strictly true as it depends on trap depth and quasi-Fermi level, as discussed in the Supporting Information, but also defect distribution. In general, SRH via band-tail states is expected to show an ideality factor less than 2, while an ideality factor of 2 is expected for states close to the middle of the gap. Therefore, on page 8 and 9, it is more accurate to say “SRH kinetics via deep defect states” rather than just SRH.

We agree with the reviewer and have changed the text accordingly on page 8, 10 and 11.

Reviewer #3:

In this communication article, Janssen et al. report a sensitive external quantum efficiency (EQE) and electroluminescence (EL) spectroscopy technique to assess radiative and non-radiative recombination losses in wide-bandgap mixed-halide lead perovskite solar cells. The authors propose a photon energy-dependent many-diode model to analyze the radiative recombination losses associated with subgap shallow defects and combine the photoluminescence quantum yield (PLQY) measurement of the quasi-Fermi level splitting limited by bulk and interface non-radiative recombination. Finally, the authors utilize devices with different perovskite compositions and choline chloride (CCI) passivation to validate the methodology.

This is overall a well-organized study with precise electrical and optical measurements and detailed modeling. However, the method proposed in this study and its conclusions are significantly important to photovoltaic research. The probe and analysis of radiative and non-radiative losses in a solar cell using EL/PLQY spectroscopy are well-established methods in the field of PV research. What the authors propose is not a sufficiently new concept or a new approach for publication in Nature Communications.

We thank the reviewer for the detailed reading of our manuscript and their thoughtful comments. Based on their comments, we have altered the manuscript to highlight the novelty and impact of our work. We have also added additional details and explanations in the main text as pointed out by the reviewer.

The paper needs more detailed discussions on how the developed method outperforms what the field knows. There is no need to use such a complex model to differentiate radiative and non-radiative recombination. The current techniques (EQE, PLQY, and ELQY) to probe radiative and nonradiative recombination are well-known and used by the field.

The motivation for the work described in this manuscript is to provide a consistent explanation for a conundrum that has been identified in recent literature in which sensitive measurements of the EQE provide a radiative voltages limit that is even less than the measured open circuit voltage (*Nat. Commun.* **11**, 5567, (2020)). A consistent explanation for this “impossible” result is missing. We outline this mystery in the introduction and, by comparing sensitive EQE and EL, we provide experimental evidence that an energy-dependent radiative ideality factor underlies this problem. This new concept leads us to formulate a many-diode model which enables including sub-bandgap states in the calculation of the radiative voltage limits in a well-founded manner.

Our model explains observations by other researchers and provides insight into the recombination kinetics of sub-bandgap states in perovskites. We do not claim that our model is better at differentiating radiative and non-radiative voltage losses compared to current techniques.

We now put additional emphasis on the importance of the many-diode model and the drawbacks of the single- and double-diode model in the introduction (page 4 and 23).

One main argument that conventional photocurrent spectroscopy measurements often overestimate the radiative open-circuit voltage (V_{oc}) loss, yielding unphysically low values of V_{oc} , is debatable. The key is that black-body radiation (BBR) from a semiconductor (Eqs. 4) cannot extend unlimitedly to zero photon energy. The shallow-defect assisted subgap radiative recombination cannot be treated simply as a band-to-band transition and contributes directly to the radiative recombination current by extending BBR integration.

The so-called single-diode model predicted V_{oc-rad} of ~ 1.15 V (Fig. 3b) could be simply estimated from the V_{oc-rad} of an ideal semiconductor with a bandgap of ~ 1.43 eV, which corresponds to the band tail of the measured real cell (Fig. 2). The V_{oc-rad} values shown in Fig. 3b and listed in Table 1 can be obtained through a simple and ideal semiconductor detail balance calculation based on corresponding bandgap values (i.e., EL peak value). The impact of subgap emission on V_{oc-rad} is negligible if a simple BBR with band cutoff is used. There is no need to run this many-diode model.

We agree that, for a first and rough approximation of V_{oc}^{rad} , estimations based on the bandgap are sufficient and there is no need to employ the full many-diode model.

Yet maximizing V_{oc}^{rad} is highly important for solar cells and pursued by many solar cell researchers, which warrants exact determination of V_{oc}^{rad} values through the measured EQE. Recent sensitive EQE measurements also record defect-response which, upon following the single diode model, give values that simply cannot be true. This can be circumvented by using a BBR cutoff, as suggested by the reviewer, but this cutoff is currently arbitrarily applied. More specifically, the BBR cutoff does not have a physical meaning and excludes measured data without a profound reason.

In our work instead, we show for the first time that defect contributions in EQE can be safely neglected, as a direct outcome of the many-diode model. Based on the reviewer's comments, we have added an additional sentence on page 13 stressing that the many-diode model enables substantiated decisions on the BBR cutoff.

We have also added an additional conclusion on page 13 where we point out that the many-diode model proves that neglecting the defect-response in EQE is a valid approach which enables using the much simpler single diode model.

We would like to address that a V_{oc}^{rad} of 1.15 V, i.e., the outcome of the single-diode model, yields an unprecedented (and unphysical) low value for the $\Delta V_{oc}^{non-rad}$ of 50 mV, and that the bandgap of the cell in Figure 2 is not defined by the onset of defect-state absorption at ~ 1.43 eV. Rather, we calculate the second derivative of the EQE to reliably determine the bandgap of this solar cell (see also Supplementary Figure 2).

On the model side, the many-diode model shown in Fig. 2a is not a physical and practical equivalent circuit model when simulating the complete J-V characteristics.

First, photon energy should not be defined as discrete segments based on the photocurrent (EQE) measurement. It does not make any physical sense when considering the J-V relationship in a diode with

a characteristic photon energy. Also, numerical interpolation should be used in the integration to solve the equation rather than a simple summation.

We agree that, ideally, the photon energy would not be discrete and instead an analytical expression for both the radiative ideality factor and the EQE would be available. However, the EQE is a measured quantity, necessarily recorded at discrete photon energies, and thereby the radiative ideality factor is also discrete.

As a compromise, we solve Equation (10) numerically after interpolating all parameters to a 3.4 meV energy spacing, yielding sufficient accuracy. Unfortunately, this interpolation was not mentioned in the text previously. We have now specified the interpolation to 3.4 meV on page 10, 11, and 12.

Furthermore, the two-diode model for radiative and non-radiative diodes is sufficient for practical device simulation. It is impractical to simulate the complete device performance for many but finite diodes. The authors can try it and will find out it is unrealistic for many diodes with R_s and R_{sh} .

We agree with the reviewer, and we now mention in the text on page 11 that the many-diode model is not practical for device simulation. Instead, our findings are predominantly of fundamental interest and provide experimental and theoretical foundation on including sensitive EQE in detailed balance calculations.

Lastly, there are insufficient new findings and significant results about the wide-bandgap mixed-halide lead perovskite solar cells.

We bring experimental evidence of largely unexplored physics of defects in semiconductors, using wide-bandgap mixed-halide perovskite solar cells as an example. We prove that the radiative ideality factor of defects is not a constant for these materials, but rather varies with defect energy with a value of ~ 2 at mid-gap energies. As a result, we formulate a many-diode model which shows that defects do not influence the V_{oc}^{rad} and which provides new insights and tools for researchers calculating/investigating radiative voltage limits (*vide supra*).

Additionally, in the final paragraph of the Results (page 21) we find that the defect response in sensitive EQE shows excellent correlation with non-radiative voltage losses in wide-bandgap mixed-halide lead perovskite solar cells (with and without top passivation). This additional new result opens a novel avenue of research and highlights the importance of using sensitive EQE to study defects in these solar cells. We now emphasize this on page 21.

REVIEWERS' COMMENTS

Reviewer #1 (Remarks to the Author):

All of the suggestions have been incorporated and reviewer comments have been addressed in the manuscript for the fulfilment of requisite quality of the journal. Hence, I recommend the publication of this work.

Reviewer #2 (Remarks to the Author):

The Authors have adequately addressed all of my comments and concerns. I recommend publication of this work in Nature Communications.

Reviewer #3 (Remarks to the Author):

The reviewer appreciates the rebuttal from the authors. After re-assessing the comments, responses, and revised manuscript, my concerns about novelty and scientific rigor have been adequately addressed. The additional data and explanations also help clarify the proposed many-diode model. I now recommend the revised manuscript for publication.

One minor correction is needed. The main text says, "Integration of the EQE with the AM1.5G spectrum affords a J_{sc} of 21.1 mA cm⁻²," which is inconsistent with the label of $J_{sc_EQE} = 20.1$ mA cm⁻² shown in Fig. 1b.

RESPONSE TO REVIEWERS COMMENTS

We thank the reviewers for their efforts in assessing the revised version of the manuscript and their positive recommendation. We are pleased to see that we have been able to resolve the questions and concerns raised previously.

Reviewer #1 (Remarks to the Author):

All of the suggestions have been incorporated and reviewer comments have been addressed in the manuscript for the fulfilment of requisite quality of the journal. Hence, I recommend the publication of this work.

Reviewer #2 (Remarks to the Author):

The Authors have adequately addressed all of my comments and concerns. I recommend publication of this work in Nature Communications.

Reviewer #3 (Remarks to the Author):

The reviewer appreciates the rebuttal from the authors. After re-assessing the comments, responses, and revised manuscript, my concerns about novelty and scientific rigor have been adequately addressed. The additional data and explanations also help clarify the proposed many-diode model. I now recommend the revised manuscript for publication.

One minor correction is needed. The main text says, "Integration of the EQE with the AM1.5G spectrum affords a J_{sc} of 21.1 mA cm⁻²," which is inconsistent with the label of $J_{sc_EQE} = 20.1$ mA cm⁻² shown in Fig. 1b.

Thank you for pointing this out. The number "21.2" was a typo it should indeed be "20.1" as reported in Fig. 1b. This has been corrected on page 6. The J_{sc} of 20.1 mA cm⁻² matches well with the 19.9 mA cm⁻² measured from the J-V scan (see bottom page 5 and top page 6).